# Clinical Utilities of Anti-Müllerian Hormone

**DOI:** 10.3390/jcm11237209

**Published:** 2022-12-04

**Authors:** Nicole Russell, Andrea Gilmore, William E. Roudebush

**Affiliations:** Department of Biomedical Sciences, University of South Carolina School of Medicine Greenville, Greenville, SC 29605, USA

**Keywords:** anti-Müllerian hormone (AMH), menopause, ovarian insufficiency, ovarian reserve, polycystic ovarian syndrome (PCOS)

## Abstract

The anti-Müllerian hormone (AMH) plays an essential role in sex determination in early embryonic development. Through a series of sequential steps that follows inheriting an XY chromosome, Sertoli cell differentiation upregulates the expression of AMH-suppressing Müllerian duct development and maintains the AMH at a high level until puberty. In females, the AMH is produced by granulosa cells of follicles beginning in the second half of fetal life and continues through adulthood, with a steady decline through the reproductive years and severe decline at menopause, until levels eventually become undetectable. The AMH is essential for the regulation of follicular maturation via the recruitment of primordial follicles throughout folliculogenesis. AMH serum concentration in women strongly correlates with ovarian reserve quantity and reflects ovulation potential. Because the AMH is expressed almost exclusively by growing follicles before FSH-dependent selection, it commonly serves as a marker for ovarian function in various clinical situations, including in the diagnosis and pathogenesis of polycystic ovarian syndrome, artificial reproductive technology, and predictions of menopause or premature ovarian failure.

## 1. Introduction

The anti-Müllerian hormone (AMH) is a member of the TGF-β super family and plays an essential role in sex determination and differentiation during embryonic development. Sex determination and differentiation involve a series of events through which an embryo must progress to acquire male or female characteristics. The default differentiation pathway is female; only when the Y chromosome is present does a male embryo develop. This sequential series of events requires a variety of genetic and hormonal input, starting with the inheritance of XX or XY chromosomes. XY chromosomes contain the SRY gene, which triggers a cascade of events resulting in a male phenotype [1]. SRY upregulates the transcription of SOX9, which initiates the testis pathway while simultaneously suppressing the ovarian pathway [1]. SOX9 triggers Sertoli cell differentiation and upregulates the expression of the AMH [1]. From the time of differentiation, Sertoli cells express a high level of AMH expression [1]. In males, Sertoli cells maintain a high level of AMH expression until puberty, after which levels decline [1]. In the presence of the AMH and androgens, internal and external genitalia will proceed to differentiate down the male pathway [1]. The first sign of male differentiation of the genital tract includes Müllerian duct regression, stimulated by a wave of apoptosis in the male fetus, which is triggered by the AMH [1]. This typically occurs around day 55 to 60 in human embryos [1]. Although initially gonadotropin independent, AMH secretion from Sertoli cells greatly decreases due to FSH stimulation by androgens and is relatively undetectable after puberty [1]. If a male fetus has low to no AMH expression, this causes the simultaneous development of male and female genitalia.

In females, the AMH is produced by granulosa cells of follicles beginning in the second half of fetal life, around the 23rd week of development, and continues through adulthood [1] with a steady decline through the reproductive years and severe decline at menopause, until levels eventually become undetectable [2]. The AMH is produced by primary to antral follicles (but not primordial follicles [1]) and has autocrine and paracrine functions as a regulator of follicular maturation [2]. After FSH-dependent selection, AMH expression is virtually undetectable, but some expression can be detected in cumulus cells of preovulatory follicles [3]. Because the AMH is expressed almost exclusively by growing follicles and before FSH-dependent selection, it commonly serves as a marker for ovarian function regarding follicular activity and maturation.

## 2. Ovarian Reserve

Females are born with a genetically predetermined number of oocytes, typically ranging between 1 and 2 million at the time of birth. The AMH is not considered essential to ovarian differentiation but inhibits Leydig cell differentiation and promotes follicular maturation [1]. The AMH is essential for the regulation of follicular maturation via the recruitment of primordial follicles throughout folliculogenesis [1]. The AMH serum concentration in women strongly correlates with ovarian reserve quantity [1] and reflects ovulation potential. Ovarian reserve is defined as the quantity and quality of primordial follicles [3]. A further distinguishment can be made between an ovarian reserve and a functional ovarian reserve (FOR). The AMH is expressed by a specific population of growing follicles, so its serum concentration does not correlate with the number of primordial follicles among young women [4]. The FOR is defined as the pool of follicles, ranging between 2 and 5 mm in diameter, from which one follicle is destined to be selected by FSH and to ovulate [3]. The follicles of the FOR are the AMH-producing follicles, which are reflected in serum AMH levels. The AMH is used as a serum marker to represent the number of growing follicles recruited from the primordial pool as an indirect measure of the ovarian reserve, since there is no direct serum marker that can be used for measurement [3]. Studies performed over the last two decades have shown that serum AMH levels have a strong correlation with the number of growing follicles and both numbers are negatively correlated with an increase in age in adult women from around age 25 [3]. AMH levels steadily increase to a peak and plateau around age 25; then, serum levels begin to decline [3]. Women without polycystic ovarian syndrome (PCOS) who have high AMH levels are commonly very fertile [5]. It is important to keep in mind that factors such as hormonal contraception and vitamin D influence the AMH serum level. The AMH serum level is lower and varies from 14 to 55% among women who use hormonal contraception [4]. Pregnancy also affects AMH levels. AMH levels decrease in the second and third trimester of pregnancy, reflecting ovarian inactivity [6]. In the first trimester of pregnancy, the concentration of the AMH stays similar to that in the time of preconception and returns to normal after birth [6]. Recent studies have shown that higher vitamin D serum concentration results in higher AMH serum levels and both can present seasonal variability during the year [5]. A vitamin D response element is present within the AMH gene promoter, thus allowing vitamin D status to influence AMH concentrations [7].

## 3. Polycystic Ovarian Syndrome

Polycystic ovarian syndrome (PCOS) is the leading cause of chronic anovulation and anovulatory infertility, due to ovarian dysfunction affecting 15–20% of reproductive-age women [8]. Symptoms vary among patients but typically include chronic anovulation, hyperandrogenism, hyperinsulinemia, and the presence of polycystic ovaries with various other with reproductive, endocrine, and/or metabolic symptoms [4]. The Rotterdam criteria require women to fulfill two of the following three criteria to be diagnosed: [5] oligo- or anovulation, [5] clinical and/or biochemical signs of hyperandrogenism, and/or [8] polycystic ovaries on ultrasound, with the exclusion of other relevant disorders [9]. Polycystic ovaries are characterized as greater than or equal to 12 follicles measuring from 2 to 9 mm in diameter, with or without the presence of increased ovarian volume that is greater than 10 mL [5]. Circulating the AMH is two- to three-fold higher in women with PCOS than in healthy women of childbearing age, most likely due to increased follicular mass and excess accumulation of pre-antral and small antral follicles with PCOS [5]. The AMH is thought to serve as a diagnostic marker and have a role in the pathogenesis of PCOS. The AMH level was also found to be positively correlated with the degree of severity of PCOS. Higher AMH levels have been shown in amenorrhoeic women compared to oligomenorrheic women with PCOS, reflecting a greater impairment in follicular development and granulosa cell function in the ovaries of amenorrhoeic women than in those of oligomenorrheic PCOS women [10]. One proposed mechanism of pathology is that the AMH inhibits the recruitment of primordial follicles out of the resting oocyte pool and reduces follicular sensitivity to FSH stimulation causing anovulation [11]. The AMH increases GnRH-dependent LH pulsatility and secretion, with the consequent dysregulation of follicle growth [12]. Previous studies have suggested that the AMH plays a direct role in the neuroendocrine dysregulation in PCOS, based on the detection of AMH receptor 2 (AMHR2) expression in hypothalamic GnRH neurons in both rodents and humans [4]. It is worth noting that there is a positive correlation between serum androgens and AMH levels via the AMH’s role in the inhibition of aromatase levels and/or activity [4]. Obesity, hyperinsulinemia, and hyperandrogenism also play major roles in the increasing AMH levels found in PCOS patients [11]. Currently, antral follicle count (AFC) via ultrasound is a criterion for the Rotterdam criteria for diagnosis; however, an evaluation of polycystic ovarian morphology (PCOM) for diagnosis of PCOS has high variability, and it can be difficult to perform an AFC trans-abdominally in virgins or obese patients [8]. It has been suggested that a more objective criterion be used for diagnosis, such as serum AMH levels, which can detect the excess of small follicles when ultrasound cannot. Serum AMH plasma levels remain stable throughout the whole menstrual cycle and are independent of the hypothalamus–pituitary axis compared to other ovarian markers, e.g., day-three FSH [8]. In a meta-analysis conducted by Iliodromiti et al., the specificity and sensitivity of serum AMH plasma levels in diagnosing PCOS in symptomatic women were 79.4 and 82.8%, respectively, for a cutoff AMH value of 4.7 ng/mL [13]. Yue et al. describe the optimal AMH diagnostic threshold for PCOS as 8.16 ng/mL in the 20–29-year-old population and 5.89 ng/mL in the 30–39-year-old population derived from data from their human studies conducted in a Chinese population of women with PCOS [5]. The recent literature shows that the AMH may also have a role in the treatment of PCOS. Serum AMH levels can predict ovarian response to clomiphene citrate (CC) or gonadotropins treatment in women with PCOS [4]. A higher AMH concentration in serum pretreatment requires higher doses of gonadotropins and extended treatment [4].

## 4. Artificial Reproductive Technology

The hormonal control of ovarian function in women struggling with fertility is influenced by the administration of exogenous FSH. The prediction of an ovarian response prior to stimulation is necessary for patient counseling and individualized tailoring of the optimal dosage of gonadotrophin for a specific patient [14]. The outcome of assisted reproductive technology (ART) is dependent on the ovarian response to ovarian stimulation, and is defined as the number and quality of eggs obtained. The ovarian response can be interpreted as the ovarian reserve. Measuring AMH levels has become a standard practice in assisted reproduction centers and is more sensitive than other methods (e.g., day-three FSH) [1]. Previous studies have shown that AMH levels may aid in the prediction of ovarian response gonadotropin stimulation protocols, with low AMH levels being correlated with a low response and a low pregnancy rate [3]. A low response to controlled ovarian hyperstimulation (COH) can be defined as the retrieval of five or less oocytes or cycle cancellation [3]. Optimum oocyte retrieval is between about 10 and 12 oocytes [15]. National guidelines suggest that the individualization of starting doses of gonadotropin for ovarian stimulation should be administered based on characteristics and markers of their ovarian reserve, including an AMH that is specific to the patient [16]. To improve responses, AMH serum levels are used, among other factors, as part of an algorithm to calculate the individual dosage of FSH for COH [3]. Under- or overstimulation can lead to cycle cancellation or ovarian hyperstimulation syndrome, so it is important that dosage is both accurate and reliable [3]. While there is no given standard among different factors regarding FSH dosing for ovarian stimulation, Nyboe Andersen et al. [17] conducted a study comparing individualized and conventional ovarian stimulation protocols in patients undergoing in vitro fertilization; the study found that individualized ovarian stimulation resulted in a higher proportion of women with 8–14 oocytes per cycle, fewer clinically relevant cases of poor or excessive ovarian response, and a less significant need for ovarian hyperstimulation syndrome preventative measures. Goswami and Nikolaou [18] found that AMH levels in patients’ serum can be used as a predictor for live births in older women undergoing IVF procedures. Women with higher AMH serum levels, greater than 21 pmol/L, had a greater live birth rate [18]. Kavoussi et al. [19] found that low serum AMH levels (described as less than 1.43 pmol/L) significantly lower the chances of blastocyst cryopreservation in patients after IVF than in those with higher AMH levels (1.5–4 ng/mL), regardless of patient age. Silbersteinet demonstrated increased implantation rates with AMH levels greater than or equal to 2.7 ng/mL, suggesting a positive correlation between AMH levels and embryo quality [4].

The prevalence of endometriosis among infertile women is particularly high. According to Coccia et al., 20–50% of women who struggle with infertility also experience endometriosis [20]. While the underlying mechanism between the pathophysiology of the disease and infertility in endometriosis is not completely understood, it is suggested that ovarian reserve is impaired through chronic inflammation, increased oxidative stress, cell cycle dysregulation, and impaired angiogenesis [21]. A study conducted by Kitajima et al. found that AMH levels in peritoneal fluid were lower in women with endometriosis compared to a control group of women without the disease [22]. Typically, endometriosis is managed with surgical innervation and medical therapies including painkiller and hormonal treatments [23]. Unfortunately, painkillers and hormonal therapies alone are unable to cure the disease and lead to high rates of symptomatic recurrence, but surgical management can further reduce ovarian reserve and cause a decrease in AMH levels. Laparoscopic surgery was considered to be a first-line gold-standard treatment for endometriosis-related infertility, with ART indicated as a second-line treatment [20]. A combined approach of both laparoscopic surgery and ART offered higher chances of pregnancy in endometriosis-related infertility; however, in cases of endometriosis, this approach could cause iatrogenic damage, leading to ovarian reserve loss and scar formation [20]. Ovarian surgery specifically lowers AMH levels and AFC, causing a decline in fertility [24]. To combat this decline in fertility, oocyte cryopreservation has been offered as preoperative fertility preservation therapy in women with endometriosis [21].

## 5. Premature Ovarian Insufficiency/Failure

Premature ovarian failure or insufficiency (POF or POI) is defined as the process of the ovaries ceasing to function before the age of 40. This early menopause is usually due to the ovaries not producing the typical amounts of estrogen or hypoestrogenism, preventing the regulated release of eggs [25]. This leads to a loss of residual follicles in the gonads and eventually infertility [26]. The overall risk of premature ovarian insufficiency before the age of 40 is 1%. The prevalence and risk of early menopause varies according to each, increasing with age: 1:10,000 in women aged 18–25; 1:1000 in women aged 25–30; and 1:100 in women aged 35–40 [27]. Diagnostically, POI is defined as a patient who is under 40 years of age, having experienced 4 months of oligomenorrhea or amenorrhea with two sets of FSH levels of >40 IU/L and an estradiol level of <50 mol/L obtained at least a month apart [28]. POI can be caused by either follicular dysfunction or follicular depletion. There are various reasons that could contribute to either cause. For example, follicular depletion could be due to a signal defect in FSH or LH receptor function, a G protein mutation, or an enzyme or autoimmune deficiency, among many other things [28]. Likewise, follicular depletion could be attributed to insufficient initial follicular count, spontaneous accelerated follicular loss, or environmentally induced follicular loss [28].

Although the exact cause of POI remains unexplained, there have been some studies showing the potential for genetic factors [26]. Ivanisevic et al. found that 15% of POI cases have a positive family history, indicating the potential for genetic etiology [27]. The genetic basis involves aberrations linked to chromosome X such as monosomy, trisomy or a translocation, or to an autosomal chromosome [27]. Likewise, Yang et al. found that the worldwide health issue of POI is due to the societal trend of delaying childbirth. They used a gene-centric approach to investigate individual gene mutations found in POI patients and decreased expression of some POI genes related to those essential for DNA repair, meiosis, or mitochondrial functions [29]. There is also recent evidence suggesting the use of cytokines as novel serum biomarkers for POI [30]. However, Rafique et al. state that nearly 90% of cases remain unexplained [28].

Kruszynska et al. suggest the role of AMH in predicting POF. AMH levels are markedly lower in women with POF, as compared to healthy individuals [31]. Additionally, Skalba et al. reported near undetectable AMH levels in POF patients [31]. This work studied 187 women, comparing the AMH plasma levels of healthy women of reproductive age to women with POF. The study found that in healthy women, the average AMH concentration was 16.97 ± 5.80 pmol/L, as compared to women with POF who had an average AMH concentration of 0.65 ± 1.81 pmol/L [31]. Likewise, Cai et al. found the average AMH serum levels in a sample of 125 patients to be 0.04 ± 0.10 ng/mL with POF and 0.69 ± 1.46 ng/mL with POI [30]. Additionally, Jiao et al. found that in 601 patients with early POI or POF, the AMH was undetectable in 75.04% of them (451/601) [21].

Zhang et al. conducted a human placenta mesenchymal stem cell (hPMSC) transplantation on granulose cell apoptosis and AMH expression in drug-induced premature ovarian failure mice [29]. They found that increasing the levels of AMH in the ovaries allowed the hPMSC transplantation to significantly improve serum levels of gonadotropins and estrogen in POF mice. This promoted follicular development and the inhibition of granulosa cell apoptosis, improving the ovarian reserve capacity [23]. Only through the upregulation of the AMH were the stem cells able to improve the state of the POF mice.

## 6. Menopause

The anti-Müllerian hormone (AMH) is a highly predictive tool for the time of menopause. Menopause, which is defined as the last menstrual period in a woman’s life, starts at the average age of 51 [20]. Menopause can range anywhere from 40 to 60 years of age. However, fertility decreases 10–13 years prior to the last official menstrual period [31]. Menopause has shown to be influenced by genetic factors, such that the age of the woman’s mother at the start of menopause may be indicative of hers. Genetic variants are known to contribute up to 50% of the variation in age at menopause [32]. However, through population-based cohort studies, Dolleman et al. found that the AMH is a more accurate predictor of the age at which menopause starts than the hereditary factor of the mother’s age [33]. A single AMH measurement can be used to predict the time of menopause even in a young woman, many years prior to the start of menstruation [31]. This has been shown through mathematical models based on a single AMH measurement and the age of the patient [34]. Tehrani et al. randomly selected 266 study participants from a pool of 1265 eligible women. Their AMH levels were measured three times at 3-year intervals. They observed reduction in AMH levels in all age groups over a 6-year period. They were able to build an accelerated failure time model using serum AMH levels at the start of follow-up to estimate the age at menopause; they correctly estimated the ages of menopause based on the different serum AMH levels among women aged 20–49 years [35]. Other studies have also been able to show the reliability of the AMH over estradiol, FSH, or inhibin B in accurately predicting menopause occurrence.

The AMH gradually increases from the time of a female’s birth, reaching its highest around the age of 25 [31]. From there, AMH levels begin to gradually decrease, until levels are nearly undetectable in post-menopausal women [31]. The AMH is more specifically a marker of non-cyclic ovarian activity. According to Ledger et al. [36], AMH levels are lowest during the luteal phase, immediately following ovulation. However, in older post-menopausal women, the range of AMH levels following a cyclic pattern was very low. Broer et al. discovered that AMH levels drop to near-undetectable levels at around 5 years prior to the start of menopause [37]. This specifically indicates the exhaustion of the stock of primordial cells. AMH serum concentrations have proven to reflect the size of the remaining antral follicle pool [37]. Additionally, parameters such as ovarian volume and the number of antral follicles indirectly indicate the time of menopause because of their association with AMH levels [31,38].

However, some sources believe that the AMH is not a precise measure of overall ovarian aging. De Kat et al. confirm that although the AMH and other ovarian reserve markers are definitely related to age at the time of menopause, they insufficiently inform the exact trajectory of an individual’s ovarian aging cycle [39]. In their prospective cohort study, Depmann et al. found that although the AMH was a significant predictor of time until menopause or TTM and time until early menopause, they found that individual predictions of age at which menopause begins had limited precision [40]. Thus, the AMH can offer a useful prediction model for younger women seeking to predict whether they will enter menopause early, but the exact age in which they will do so has room for error. Another factor to consider is that with increasing age, the predictive capacity of the AMH decreases.

## 7. Conclusions

The AMH is a transforming growth factor hormone that is essential in growth, sexual differentiation, and folliculogenesis. In females, the AMH is produced by granulose cells from fetal life, throughout adulthood and then steadily declines until reaching menopause. There is a strong correlation between the AMH and ovulation potential and ovarian reserve quantity, making it a reliable marker for ovarian function. Serum AMH levels contribute to clinical scenarios and diagnoses relating to polycystic ovarian syndrome (PCOS), artificial reproductive technology (ART), risk of premature ovarian failure (POF), and menopause timeline prediction. The AMH is an essential tool in predicting ovarian function and thus reproductive lifespan, which provides needed knowledge in a variety of clinical scenarios to better assist in the treatment of patients.

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
