# Peer review of "Clinical Utilities of Anti-Müllerian Hormone"

_jcm, 2022, doi:10.3390/jcm11237209_

Round 1

Reviewer 1 Report

The authors of review entitled “Clinical Utilities of Anti-Müllerian Hormone” evaluated the role of Anti-Müllerian Hormone (AMH) in pathogenesis, diagnosis and treatment of various pathologies related to reproductive system. The review implied that AMH had role in in sex determination in early embryonic development, and serves as a marker for ovarian function in various clinical situations including diagnosis and pathogenesis of polycystic ovarian syndrome, artificial reproductive technology, and predictions of menopause or premature ovarian failure, which puts in light the importance of present manuscript and its significance to wide range of readers.

However, there are few inconsistences, which should be correct.

First, there are some typographic and grammar issues to correct throughout the text.

Introduction section was well written, with appropriate background of investigated issues.

Please be more detailed regarding PCOS and AMH. Include more studies considering the role of AMH in PCOS in animal and human studies.

The reference list was not written consistently. Please correct.

Author Response

  1. There are some typographic and grammar issues to correct throughout the text.
    1. Manuscript reviewed and corrected.
  2. The reference list was not written consistently. Please correct.
    1. Reference list was reformatted
  3. Please be more detailed regarding PCOS and AMH. Include more studies considering the role of AMH in PCOS in animal and human studies.
    1. More human and animal study information added to the PCOS section.

Reviewer 2 Report

There is a spelling mistake on line 90:  "chorionic" should be replaced by "chronic". The content is well structured, very well updated as information and very useful on the subject.

Author Response

  1. There is a spelling mistake on line 90:  "chorionic" should be replaced by "chronic".
    1. Chorionic is corrected to chronic.

Reviewer 3 Report

In this narrative manuscript , the authors described the clinical use of AMH.

The manuscript is well written and of interest for the readers 

There are some some missing informations

- In the PCOS , the number of antral follicles should be precised on P 3 line 97

- In the different clinical situations , it should be interesting to find mean  measurements of AMH and specially in ART and POI.

- POI should be discussed before menopause.

- No information is given for endometriotic patients even in this group of patients , the AMH level is very interesting in the clinical management of patients . What are the indications of oocytes cryopreservation in endometriotic patients with AMH level < 1ng/ml ?

Some data coud be found in manuscripts previously published in JCM.

There is no conclusion nor any take home message.

Author Response

  1. In the PCOS , the number of antral follicles should be precised on P 3 line 97
    1. The number and size of follicles was added to P3 line 97.
  2. In the different clinical situations , it should be interesting to find mean  measurements of AMH and specially in ART and POI.
    1. More values were added to the ART and POI sections.
  3. POI should be discussed before menopause.
    1. Order of manuscript was changed
  4. No information is given for endometriotic patients even in this group of patients , the AMH level is very interesting in the clinical management of patients . What are the indications of oocytes cryopreservation in endometriotic patients with AMH level < 1ng/ml ?
    1. Information regarding endometriosis and cryopreservation was added in the ART section of the manuscript.
  5. Some data coud be found in manuscripts previously published in JCM.
    1. Manuscript was checked via “plagiarism checker” provided by Baer Yang at JCM.
  6. There is no conclusion nor any take home message.
    1. Conclusion was added.